# Antibiotic resistance and serotype distribution of *Shigella* strains in Bangladesh over the period of 2014–2022: evidence from a nationwide hospital-based surveillance for cholera and other diarrheal diseases

Mokibul Hassan Afrad,[1] Md. Taufiqul Islam,[1] Yasmin Ara Begum,[1] Md. Saifullah,[1] Faisal Ahmmed,[1] Zahid Hasan Khan,[1] Zakir Hossain Habib,[2] Ahmed Nawsher Alam,[2] Tahmina Shirin,[2] Taufiqur Rahman Bhuiyan,[1] Edward T. Ryan,[3] Ashraful Islam Khan,[1] Firdausi Qadri[1]

**ABSTRACT** The objective of this study was to assess the prevalence, antimicrobial resistance patterns, and risk factors linked to *Shigella* infections through a nationwide hospital-based diarrheal disease surveillance in Bangladesh. From May 2014 to May 2022, from a systematic sentinel surveillance of *Shigella* infections in over 10 hospitals across Bangladesh, stool specimens were collected from patients with acute watery diarrhea and tested for *Shigella* species by microbiological culture. The susceptibility to antibiotics was tested using the disk diffusion method. Structured questionnaires were used to collect participants' socioeconomic status and clinical, sanitation, and food history. Out of 24,357 stool specimens, 1.8% were positive for *Shigella* species, with a higher prevalence among males (58%). Children in the 6–17 age group were found to be at the highest risk of *Shigella* infections. The most prevalent serotype was *Shigella flexneri* (79.5%), followed by *Shigella sonnei*. *S. flexneri* serotype 2a was the most common (63.3%) among all *Shigella* serotypes. Antibiotic susceptibility testing showed over 99% of isolates resistant or with intermediate susceptibility to one of the seven antibiotics tested. About 96% of *S. flexneri* and all *S. sonnei* isolates demonstrated resistance to at least one quinolone class of antibiotics, particularly ciprofloxacin or nalidixic acid. *S. sonnei* showed higher antibiotic resistance and multidrug resistance compared to *S. flexneri*. The high level of resistance to ciprofloxacin highlights the need for more prudent use of this antibiotic and improved hygiene and sanitation. The study emphasized the importance of regular monitoring of drug resistance to effectively manage *Shigella* infections. These findings may provide the epidemiological evidence for conducting future appropriate *Shigella* vaccine clinical trials in Bangladesh.

**IMPORTANCE** This nationwide study in Bangladesh assessed *Shigella* infections from 2014 to 2022 from clinical samples. *S. flexneri* was predominant, with concerning antibiotic resistance, notably to ciprofloxacin and nalidixic acid in over 96% of isolates. This emphasizes the urgency of prudent antibiotic use and improved hygiene. The findings provide crucial antimicrobial resistance patterns of *Shigella* species, highlighting the need for ongoing resistance monitoring and potentially informing future vaccine trials.

**KEYWORDS** diarrhea, *Shigella*, prevalence, antibiotic resistance, nationwide surveillance, Bangladesh

*S*higella spp. are a major cause of acute watery diarrhea leading to complications particularly in low-income countries (LMIC )where healthcare facilities, water and

Address correspondence to Firdausi Qadri, fqadri@icddrb.org.

Ashraful Islam Khan and Firdausi Qadri contributed equally to this article.

The authors declare no conflict of interest.

See the funding table on p. 11.

sanitation systems, and treatment options are limited (1). In 2016, *Shigella* was estimated to cause 2.1 million deaths, accounting for around 13.2% of all diarrhea-related deaths (2). The Global Enteric Multicenter Study (GEMS) suggested that the burden of *Shigella* may be twice as high as previously estimated, ranking it as the most commonly detected pathogen (3).

The genus *Shigella* is composed of four species, including *Shigella flexneri*, *Shigella sonnei*, *Shigella dysenteriae*, and *Shigella boydii*, which differ based on the O antigen present on lipopolysaccharide walls (4). The four *Shigella* species and their different serotypes vary in their geographical distribution and epidemiological significance. Currently, the two most common species causing shigellosis are *S. flexneri* and *S. sonnei*, with *S. flexneri* being more prevalent in Asia and Africa and *S. sonnei* being more common in high-income countries (5, 6). However, the distribution of *S. sonnei* is increasing globally, including in LMICs) and is replacing *S. flexneri* as the predominant species (5–7).

The World Health Organization (WHO) advises that all instances of bloody diarrhea be treated immediately with an antimicrobial effective against *Shigella* to minimize the risk of complications, shorten the duration of illness, and prevent transmission to others (8). However, the primary challenge in treating *Shigella* is the emergence of multidrug-resistant strains, including growing resistance to third-generation cephalosporins, fluoroquinolones, and azithromycin (AZM) (9, 10). Fluoroquinolones such as ciprofloxacin (CIP) are currently the recommended first-line treatment for shigellosis, but in Asian countries like Bangladesh, the effectiveness of this antibiotic has been compromised by the emergence of resistant *Shigella* varieties (9, 11, 12).

Here, we present data from the nationwide hospital-based foodborne illness surveillance, where monitoring of *Shigella* antibiotic resistance has been ongoing since 2014. Additionally, we discuss the distribution of *Shigella* serotypes and antibiotic resistance pattern over a 9-year span in Bangladesh.

## MATERIALS AND METHODS

### Study site and population

In May 2014, the International Centre for Diarrhoeal Disease Research, Bangladesh (icddr,b), and the Institute of Epidemiology, Disease Control and Research (IEDCR), Bangladesh, collaboratively started the enteric food-borne illness surveillance in 10 hospitals in eight different districts in Bangladesh (Fig. 1). However, there was a gap in funding which resulted in the interruption of the surveillance from January to May of 2016. The 10 sites were selected based on reports of acute watery diarrhea and previous surveillance studies, from the Directorate General of Health Services and other sources (13, 14). In this study, participants were enrolled from May 2014 to May 2022.

### Surveillance

The surveillance methods utilized in this study have been previously described in Khan et al. (15). The procedure has been summarized below for the reader's convenience.

Surveillance covered all age groups, but distinct case definitions were applied for those under 2 months and those 2 months or older: (i) for infants under 2 months, changed stool habit from usual pattern in terms of frequency (more than the usual number of purgings) or nature of stool (more water than fecal matter); (ii) for individuals aged 2 months or older, diarrhea was defined as any patient presenting at the hospital with three or more loose or liquid stools within 24 h, or with three or fewer loose/liquid stools leading to dehydration within the last 24 h.

At each surveillance site, a team including a physician, nurse, medical technologist, and trained field attendant was established. At each site, a daily list of diarrhea patients was prepared (both inpatient and outpatient). Four patients meeting the case definition, without severe comorbidities (e.g., severe acute respiratory illness, acute cardiovascular symptoms, or severe acute neurological disorder), were enrolled each day from Saturday

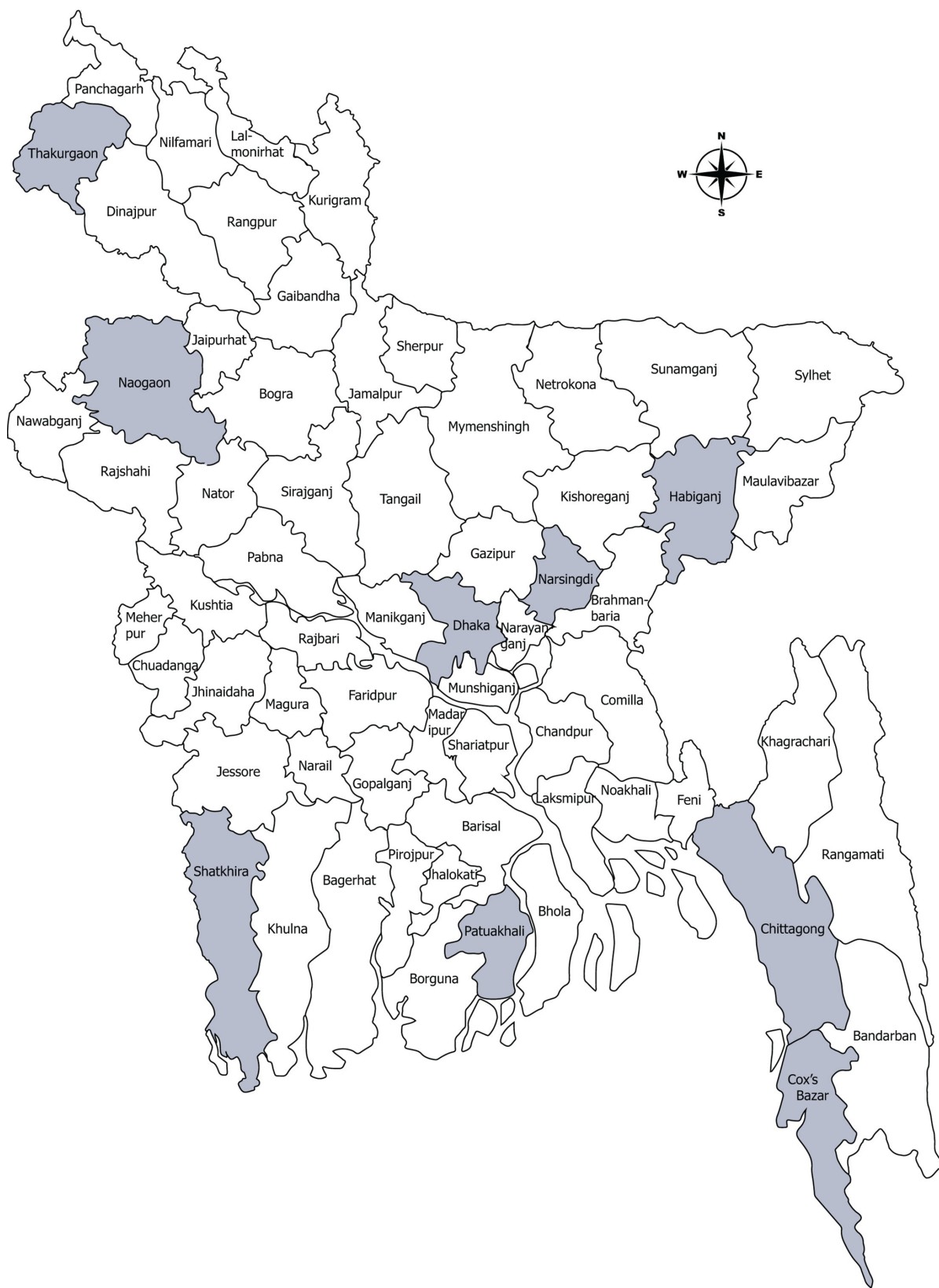

**FIG 1**   Map of nationwide study surveillance sites in Bangladesh, May 2014–May 2022. The map was created by ArcGIS Pro version 3.2.

to Wednesday. Two patients under 5 years old and two patients aged 5 years or older were enrolled each day. If the target number was not met in one age group, patients from the other group were included to meet the target of four patients. Upon receiving consent, the physician collected sociodemographic characteristics, such as age, gender, profession, diet history, medical history, sanitation, and hygiene information, and a stool sample for testing. The specimens were collected in Cary-Blair transport media and buffered glycerol saline (BGS) media and transported within 15 days to the laboratories in Dhaka at the icddr,b, and IEDCR, following cold-chain procedures (2°C–8°C), for immediate laboratory processing.

## Laboratory procedure

The conventional microbiological culture was carried out from both Carry-Blair and BGS independently by streaking directly on *Salmonella Shigella* (SS) agar (Becton Dickinson, France) and incubated at 37°C for 16–22 h. The resultant *Shigella*-like colonies were picked onto Kligler iron agar, motility indole urea, and Simmon citrate agar (Becton Dickinson) and then analyzed further through biochemical tests to identify and isolate any potential *Shigella* colonies. The serotyping of *Shigella* isolates was carried out using *Shigella* Antisera (Denka Seiken, Tokyo, Japan) and monoclonal antibody reagents (MASF IV-1 and MASF IV-2; Reagensia AB, Stockholm, Sweden).

## Antimicrobial susceptibility testing

Susceptibility to antimicrobial agents, including ampicillin (AMP, 10 µg), AZM (15 µg), CIP (5 µg), ceftriaxone (CRO, 30 µg), mecillinam (MEL, 10 µg), nalidixic acid (NA, 30 µg), and sulfomethoxazole-trimethoprim (SXT, 1.25/23.75 µg) (Oxoid, Basingstoke, United Kingdom), was determined by the disk diffusion method, which was recommended by the Clinical and Laboratory Standards Institute (16). For *S. flexneri*, the susceptibility of every fifth positive isolate was tested ($n = 185$), while for *S. sonnei* ($n = 68$), all of the isolates were tested. *Escherichia coli* American Type Culture Collection 25922 strain was used as the susceptible control strain.

## Statistical analysis

The data were entered into Microsoft Structured Query Language server for analysis. The demographics, clinical characteristics, and behavioral components were summarized using proportions for categorical variables and median with interquartile range for continuous variables. Bivariate analysis using Pearson's chi-squared test was performed for the primary analysis to determine any statistical differences, with a significance level of $P \le 0.05$. Logistic regression was used to identify the predictors of enteric diseases that fit in the model with a significance level of $P < 0.05$ from the chi-squared test, and the results were expressed as crude and adjusted odds ratios.

## RESULTS

From May 2014 to May 2022, a total of 31,517 patients were enrolled in the surveillance (Table 1), with 55.7% being male. Over 50% of the patients with diarrhea enrolled were under five years old, 30.3% were between 18 and 45 years old, and 7% were between 46 and 60 years old (Table 2). About 20% of the enrolled patients were housewives and 6.7% were service holders (e.g. teachers, bankers, and public servants). Most patients reported some level (61.5%) or severe (12.7%) of dehydration. Other reported symptoms included vomiting (63.4%), fever (53.5%), and abdominal cramps (50.7%). 25% of the patients had consumed food from roadside vendors, 82.6% had drunk untreated water, and 20.8% had neighbors with diarrhea in the week prior to their illness.

Out of the 31,517 patients enrolled, 24,357 stool specimens were available for microbiological culture, and 1.8% (445/24,357) of these tested positive for *Shigella* spp (Table 1). The majority of the positive cases were under 5 years old and 27% were 18–45 years old, 58% were male (Table 2). In these cases, majority of the cases reported some (70%) or severe (5.4%) dehydration, 58% had vomiting, 62% had abdominal cramps, and

**TABLE 1** Site-wise distribution of diarrhea samples and *Shigella*-positive organisms from May 2014 to May 2022

| Division | Surveillance sites | Participants enrolled | Culture performed | *Shigella* positive, *n* (%) | |
|---|---|---|---|---|---|
| | | | | Site-wise | Division-wise |
| Dhaka | Narsingdi | 1,745 | 1,737 | 32 (1.8) | |
| | DMCH,[a] South Dhaka | 1,960 | 1,453 | 30 (2.1) | |
| | UAMC&H,[b] North Dhaka | 1,539 | 1,036 | 35 (3.4) | 97 (2.3) |
| Chattogram | Cox's Bazar | 4,385 | 3,298 | 71 (2.2) | |
| | BITID[c] | 3,911 | 2,852 | 56 (2) | 127 (2) |
| Sylhet | Habiganj | 4,816 | 3,564 | 71 (2) | 71 (2) |
| Rajshahi | Naogaon | 3,570 | 2,601 | 50 (1.9) | 50 (1.9) |
| Barisal | Patuakhali | 3,946 | 2,930 | 52 (1.8) | 52 (1.8) |
| Rangpur | Thakurgaon | 2,439 | 2,417 | 7 (0.3) | 7 (0.3) |
| Khulna | Shatkhira | 3,206 | 2,469 | 41 (1.7) | 41 (1.7) |
| Total | | 31,517 | 24,357 | 445 (1.8) | 445 (1.8) |

[a]DMCH, Dhaka Medical College Hospital.
[b]UAMC&H, Uttara Adhunik Medical College and Hospital.
[c]BITID, Bangladesh Institute of Tropical and Infection Diseases.

60% reported purging rates of more than 10 times in the last 24 h. About, 28% of those positive had consumed food from street vendors, 82% had drunk untreated water, and 19.8% had neighbors with diarrhea in the week before their illness.

We found that the risk of *Shigella* infection was lower among patients with diarrhea who were younger than 6 years old compared to other age groups (Table 3). The risk was 2.4 times higher in those between 6 and 17 years old (95% CI: 1.57–3.78) and 1.96 times higher in those over 61 years old (95% CI: 1.01–3.8) compared to those under 6. Additionally, patients infected with *Shigella* had an 11.41 times higher risk (95% CI: 8.21–15.85) of having blood in their stool.

The isolation rate of *Shigella* varied from 0.3% to 3.4% (Table 1), and phenotypic identification revealed that 79.5% were *S. flexneri*, 18.8% were *S. sonnei*, 1.2% were *S. boydii*, and 0.5% were *S. dysenteriae* (Table 4). Throughout the study, *S. flexneri* remained the most prevalent strain. The isolation rate of *S. sonnei* increased over the years, with the highest rate seen in 2021 (33.3%). The most prevalent *S. flexneri* serotype was 2 a (79.7%) followed by 3 a (10.8%). Six (1.8%) *S. flexneri* isolates were not able to be typed. For *S. sonnei*, phase-I serotype was the most common from 2014 to 2019.

### *Shigella* AMR in Bangladesh

Out of the 253 *S. flexneri* and *S. sonnei* isolates tested for antibiotic sensitivity, 98.8% (250 of 253) were either resistant or had intermediate susceptibility to at least one of the seven antibiotics tested (Table 5). Of the 185 *S. flexneri* isolates, 96% were found to be resistant to at least one of the quinolone class of antibiotics, such as CIP or NA.

Of the *S. flexneri* isolates, 94% were found to have either intermediate resistance or resistance to NA, followed by CIP (84.4%), SXT (46%), ampicillin (46%), AZM (23.8%), MEL (7.1%), and CRO (3.3%). Meanwhile, *S. sonnei* isolates showed higher resistance rates to a variety of antibiotics, including 100% resistance to NA, followed by 98.5% to CIP, 83.8% to AZM, 73.5% to SXT, 66.2% to AMP, and 20.6% to CRO.

Out of the 253 *Shigella* isolates tested for antibiotic resistance, 29 different resistance patterns were identified (Table 5). Out of these, 69% (20 out of 29) were multidrug resistant (MDR), meaning they were resistant to at least three classes of antibiotics. Of the total 253 isolates, 175 (69%) were MDR, with 112 out of 182 (61.5%) *S. flexneri* isolates and 60 out of 68 (88.2%) *S. sonnei* isolates being MDR. The study found that *S. sonnei* had a higher frequency of MDR compared to *S. flexneri* isolates.

**TABLE 2** Factors associated with *Shigella*

| Characteristics | N = 24,357[a] | Negative, n = 23,912[a] | Positive, n = 445[a] | P value[b] |
|---|---|---|---|---|
| Age group (years) | | | | **<0.001** |
| 0–5 | 12,777 (52.5) | 12,588 (98.5) | 189 (1.5) | |
| 6–17 | 986 (4.0) | 941 (95.4) | 45 (4.6) | |
| 18–45 | 7,372 (30.3) | 7,250 (98.3) | 122 (1.7) | |
| 46–60 | 1,670 (6.9) | 1,637 (98.0) | 33 (2.0) | |
| ≥61 | 1,552 (6.4) | 1,496 (96.4) | 56 (3.6) | |
| Age in years | 3 (0, 35) | 3 (0, 35) | 14 (2, 40) | **<0.001** |
| Gender | | | | 0.300 |
| Female | 10,797 (44.3) | 10,611 (98.3) | 186 (1.7) | |
| Male | 13,560 (55.7) | 13,301 (98.1) | 259 (1.9) | |
| Occupation | | | | **<0.001** |
| Service holder | 1,635 (6.7) | 1,611 (98.5) | 24 (1.5) | |
| Housewife | 4,772 (19.6) | 4,678 (98.0) | 94 (2.0) | |
| Agriculture worker | 830 (3.4) | 810 (97.6) | 20 (2.4) | |
| Businessman | 979 (4.0) | 963 (98.4) | 16 (1.6) | |
| Labor/worker/driver | 800 (3.3) | 782 (97.8) | 18 (2.2) | |
| Student and unemployed | 1,986 (8.2) | 1,924 (96.9) | 62 (3.1) | |
| Child (up to 10 years) | 13,149 (54.0) | 12,941 (98.4) | 208 (1.6) | |
| Others | 206 (0.8) | 203 (98.5) | 3 (1.5) | |
| Duration of diarrhea (days) | | | | 0.009 |
| <3 | 12,708 (52.2) | 12,444 (97.9) | 264 (2.1) | |
| 3–7 | 11,463 (47.1) | 11,285 (98.4) | 178 (1.6) | |
| ≥8 | 186 (0.8) | 183 (98.4) | 3 (1.6) | |
| Number of purging in last 24 h | | | | 0.400 |
| ≤10, times | 9,248 (38.0) | 9,071 (98.1) | 177 (1.9) | |
| >10, times | 15,109 (62.0) | 14,841 (98.2) | 268 (1.8) | |
| Nature of stool | | | | **<0.001** |
| Loose watery | 13,159 (54.0) | 12,918 (98.2) | 241 (1.8) | |
| Rice watery | 10,101 (41.5) | 9,945 (98.5) | 156 (1.5) | |
| Bloody | 121 (0.5) | 91 (75.2) | 30 (24.8) | |
| Formed | 976 (4.0) | 958 (98.2) | 18 (1.8) | |
| Vomiting | | | | 0.021 |
| No | 8,906 (36.6) | 8,720 (97.9) | 186 (2.1) | |
| Yes | 15,451 (63.4) | 15,192 (98.3) | 259 (1.7) | |
| Dehydration | | | | **<0.001** |
| No | 6,277 (25.8) | 6,168 (98.3) | 109 (1.7) | |
| Some | 14,991 (61.5) | 14,679 (97.9) | 312 (2.1) | |
| Severe | 3,089 (12.7) | 3,065 (99.2) | 24 (0.8) | |
| Abdominal cramp | | | | **<0.001** |
| No | 12,020 (49.3) | 11,850 (98.6) | 170 (1.4) | |
| Yes | 12,337 (50.7) | 12,062 (97.8) | 275 (2.2) | |
| Fever | | | | **<0.001** |
| No | 11,318 (46.5) | 11,147 (98.5) | 171 (1.5) | |
| Yes | 13,039 (53.5) | 12,765 (97.9) | 274 (2.1) | |
| Tap water | | | | 0.023 |
| No | 18,155 (74.5) | 17,844 (98.3) | 311 (1.7) | |
| Yes | 6,202 (25.5) | 6,068 (97.8) | 134 (2.2) | |
| Tube well | | | | 0.026 |
| No | 4,134 (17.0) | 4,041 (97.8) | 93 (2.2) | |
| Yes | 20,223 (83.0) | 19,871 (98.3) | 352 (1.7) | |
| Bottled water | | | | 0.300 |
| No | 21,428 (88.0) | 21,030 (98.1) | 398 (1.9) | |

**TABLE 2** Factors associated with *Shigella* (*Continued*)

| Characteristics | $N = 24{,}357^a$ | Negative, $n = 23{,}912^a$ | Positive, $n = 445^a$ | *P* value[b] |
|---|---|---|---|---|
| Yes | 2,929 (12.0) | 2,882 (98.4) | 47 (1.6) | |
| Water treated by boiled/filtered/chemical | | | | 0.800 |
| No | 20,128 (82.6) | 19,762 (98.2) | 366 (1.8) | |
| Yes | 4,229 (17.4) | 4,150 (98.1) | 79 (1.9) | |
| Take food from roadside | | | | 0.200 |
| No | 18,209 (74.8) | 17,888 (98.2) | 321 (1.8) | |
| Yes | 6,148 (25.2) | 6,024 (98.0) | 124 (2.0) | |
| Take food from large gatherings | | | | 0.200 |
| No | 21,203 (87.1) | 20,825 (98.2) | 378 (1.8) | |
| Yes | 3,154 (12.9) | 3,087 (97.9) | 67 (2.1) | |
| Any one of neighbors has the same disease | | | | 0.600 |
| No | 19,291 (79.2) | 18,934 (98.1) | 357 (1.9) | |
| Yes | 5,066 (20.8) | 4,978 (98.3) | 88 (1.7) | |

[a]*n* (%), median (interquartile range).
[b]Pearson's chi-squared test, Wilcoxon's rank-sum test, Fisher's exact test. Statistically significant values (*P* < 0.05) are highlighted in bold text.

## DISCUSSION

This study presents data on the prevalence and types of *Shigella* spp. isolated in patients with acute diarrhea and the clinical characteristics and risk factors associated with *Shigella* infections in Bangladesh.

We found that *Shigella* is prevalent throughout the country, and its incidence has remained stable over the course of the study, with a range from from 0.3% to 3.4% (Table 1). Division-wise (administrative area), except Rangpur Division, all other administrative divisions exhibited a similar *Shigella* detection rate ranging from 1.7% to 2.3%, whereas Rangpur Division recorded the lowest detection rate at 0.3% (Table 1).

*Shigella* infection was seen in people of all ages, but children under 5 years old had the most cases. The highest positive detection rate was observed in children aged 6–17 years. Males were more likely to be infected with *Shigella*, with 58% of the cases being male, which aligns with previous studies by Khan et al. and Taneja (17, 18).

The study found that *S. flexneri* was the most prevalent species of the *Shigella* genus, accounting for 80% of cases, which is in line with other studies conducted in Bangladesh and developing countries in Africa and Asia (19). However, there was a significant increase in the prevalence of *S. sonnei* cases, rising from 10% in 2014 to 24.4% in 2017. Recent evidence suggests that *S. sonnei* is emerging as the dominant cause of shigellosis in countries undergoing economic transition (20), and this study showed a similar trend.

Recently, *S. sonnei* has become more prevalent in certain regions of Bangladesh, such as Shatkhira district, where it was found in 78.3% of cases. Over the years, *S. sonnei* has become more frequent in Bangladesh, from 12% in 2004 to 25% in 2011 in urban areas (14), and from 35% in 2010 to 41% in 2012 in rural areas (7, 21). Such increase has also been documented in other Asian countries such as India (22), Pakistan (23), Vietnam (24), and Thailand (25). The GEMS has recognized this trend and recommended the development of a quadrivalent vaccine to protect against *S. sonnei* and *S. flexneri* (serotypes 2a, 3a, and 6) in endemic regions (19). Research has linked this increase in *S. sonnei* to economic growth (20). As Bangladesh continues to develop economically and improve its sanitation, *S. sonnei* may become a more significant public health concern in the future.

The study found that *S. flexneri* 2a was still the most prevalent serotype in Bangladesh, consistent with previous research. Seven subtypes of *S. flexneri* were identified; serotype 2a was the most widespread subtype, accounting for 63% of *Shigella* cases. Additionally, atypical serotypes such as 1a, 1b, 1c, 4a, and 6 were also detected.

Antimicrobial resistance has been the key driver of the evolution of *Shigella* species, especially in the LMICs. This has been attributed to factors such as over-prescription and

**TABLE 3** Clinical data and risk factors of stool culture-positive *Shigella* [using generalized linear mixed-effect models (random: surveillance sites)][a,b]

| Characteristics | Labels | cOR (95% CI) | *P* value[c] | aOR (95% CI) | *P* value |
|---|---|---|---|---|---|
| Age (years) | <6 (ref) | 1.00 | | | |
| | 6–17 | 2.94 (2.09–4.13) | **<0.001** | 2.44 (1.57–3.78) | **<0.001** |
| | 18–45 | 1.02 (0.8–1.3) | 0.879 | 0.98 (0.5–1.9) | 0.95 |
| | 46–60 | 1.23 (0.84–1.8) | 0.282 | 1.14 (0.56–2.35) | 0.714 |
| | ≥61 | 2.29 (1.67–3.14) | **<0.001** | 1.96 (1.01–3.8) | **0.047** |
| Occupation | Service holder (ref) | 1.00 | | | |
| | Housewife | 1.43 (0.9–2.27) | 0.126 | 1.39 (0.88–2.17) | 0.155 |
| | Agriculture worker | 1.86 (1.0–3.43) | **0.048** | 1.62 (0.89–2.96) | 0.115 |
| | Businessman | 1.16 (0.61–2.21) | 0.649 | 1.15 (0.61–2.14) | 0.664 |
| | Labor/worker/driver | 1.63 (0.88–3.02) | 0.124 | 1.64 (0.9–2.99) | 0.103 |
| | Student and unemployed | 2.27 (1.4– 3.68) | **0.001** | 1.5 (0.9–2.5) | 0.119 |
| | Child (up to 10 years) | 1.28 (0.82–2.01) | 0.277 | 1.47 (0.72–3.01) | 0.292 |
| | Others | 1.01 (0.29–3.47) | 0.986 | 0.83 (0.25–2.7) | 0.753 |
| Duration of diarrhea (days) | <3 (ref) | 1.00 | | | |
| | 3–7 | 0.84 (0.69–1.04) | 0.105 | 0.79 (0.64–0.97) | **0.022** |
| | ≥8 | 0.81 (0.25–2.56) | 0.715 | 0.63 (0.21–1.86) | 0.399 |
| Nature of stool | Loose watery (ref) | 1.00 | | | |
| | Rice watery stool | 0.83 (0.64–1.06) | 0.131 | 0.85 (0.66–1.08) | 0.184 |
| | Bloody | 17.08 (11.02–26.48) | **<0.001** | 11.41 (8.21–15.85) | **<0.001** |
| | Formed | 0.93 (0.56–1.55) | 0.793 | 0.88 (0.54–1.43) | 0.601 |
| Vomiting | Yes | 0.76 (0.61–0.93) | **0.010** | 0.72 (0.59–0.88) | **0.002** |
| Dehydration | None | 1.00 | | | |
| | Some | 1.2 (0.94–1.52) | 0.148 | 1.13 (0.9–1.43) | 0.301 |
| | Severe | 0.56 (0.35–0.91) | **0.018** | 0.55 (0.34–0.89) | **0.014** |
| Abdominal cramp | Yes | 1.38 (1.11–1.7) | **0.003** | 1.18 (0.93–1.51) | 0.176 |
| Fever | Yes | 1.44 (1.17–1.78) | **0.001** | 1.42 (1.16–1.73) | **0.001** |
| Tap water | Yes | 1.03 (0.8–1.32) | 0.818 | 1.09 (0.82–1.45) | 0.562 |
| Tube well | Yes | 0.91 (0.67–1.24) | 0.558 | 0.95 (0.69–1.31) | 0.751 |

[a]Note: all no exposures as reference.
[b]aOR, adjusted odds ratio; cOR, crude odds ratio.
[c]P values ≤0.05 are presented in bold.

easy access to antibiotics (26, 27). The study found that 98.8% of the tested *Shigella* isolates in Bangladesh showed resistance or intermediate susceptibility to at least one of the seven antibiotics, and 41.8% had MDR profiles (Tables 5 and 6). There were differences in the antibiotic resistance profile between *S. flexneri* and *S. sonnei*, with the latter having a higher MDR level of 92.6%. Sixty-six percent of *S. sonnei* strains were resistant to commonly used antibiotics such as AMP, AZM, or CIP, while the dominant resistance profile among *S. flexneri* was AMP-CIP-NA-SXT (81.8%). A previous study in Bangladesh documented that 95% of *S. sonnei* and 66% of *S. flexneri* had MDR profiles, and a small number of isolates were found to be extensively drug-resistant (28).

The study found differences in the AMR profiles between *S. flexneri* and *S. sonnei*. According to the WHO, CIP is recommended as the first-line treatment for shigellosis (27), a high percentage of both *S. flexneri* and *S. sonnei* strains showed resistance or intermediate resistance to CIP, with resistance rates of 77% and 98%, respectively ()(Table 5). Among previous studies conducted in Bangladesh, Azmi et al. also reported a gradual increase in resistance of CIP from 0% in 2004 to 44% in 2010 (29). Ud-Din et al. also showed evidence of a dramatic change of resistance of *S. sonnei* to CIP from 10% in 2007 to 70% in 2011 (7). Similar trends have been observed in other South Asian countries like India, Nepal, and China (30, 31). The *S. sonnei* strains also showed a high resistance (66.2%) to AMP, while *S. flexneri* showed 46% resistance to AMP.

Previous studies from Southeast Asia reported the resistance of *Shigella* spp. to cephalosporins at 2.0%–5.2% (22, 30). In this study, we found a higher resistance to cephalosporins, with 20.6% of *S. sonnei* isolates showing resistance and 36.8% showing

**TABLE 4** *Shigella* serotypes in Bangladesh from May 2014 to May 2022

| *Shigella* spp. | 2014 | 2015 | 2016[b] | 2017 | 2018 | 2019 | 2020 | 2021 | 2022 | Grand total |
|---|---|---|---|---|---|---|---|---|---|---|
| | n (%) | n (%) | n (%) | n (%) | n (%) | n (%) | n (%) | n (%) | n (%) | N (%) |
| *S. flexneri*[a] | 26 (86.7) | 46 (100) | 26 (86.7) | 62 (75.6) | 45 (84.9) | 37 (72.5) | 35 (76.1) | 34 (66.7) | 23 (74.2) | 334 (79.5) |
| 6 | | | | 1 (1.6) | | | | | | 1 (0.3) |
| 1a | | | | 1 (1.6) | | | | | | 1 (0.3) |
| 1b | | | | | 3 (6.7) | | | | | 3 (0.9) |
| 1c | | 1 (2.2) | 1 (3.8) | 3 (4.8) | | 1 (2.7) | | | | 6 (1.8) |
| 2a | 15 (57.7) | 37 (80.4) | 19 (73.1) | 45 (72.6) | 36 (80.0) | 34 (91.9) | 34 (97.1) | 28 (82.4) | 18 (78.3) | 266 (79.6) |
| 3a | 9 (34.6) | 3 (6.5) | 3 (11.5) | 10 (16.1) | 4 (8.9) | | 1 (2.9) | 4 (11.8) | 2 (8.7) | 36 (10.8) |
| 4a | | 2 (4.3) | 3 (11.5) | 1 (1.6) | 1 (2.2) | 1 (2.7) | | | | 8 (2.4) |
| X | | | | 1 (1.6) | | | | 1 (2.9) | 3 (13.0) | 5 (1.5) |
| Y | 1 (3.8) | | | | | 1 (2.7) | | | | 2 (0.6) |
| Untypeable | 1 (3.8) | 3 (6.5) | | | 1 (2.2) | | | 1 (2.9) | | 6 (1.8) |
| *S. sonnei* | 3 (10.0) | | 3 (10.0) | 20 (24.4) | 5 (9.4) | 13 (25.5) | 11 (23.9) | 17 (33.3) | 7 (22.6) | 79 (18.8) |
| Phase I | 3 (100.0) | | 2 (66.7) | 11 (55.0) | 3 (60.0) | 7 (53.8) | 4 (36.4) | 2 (11.8) | 1 (14.3) | 33 (41.8) |
| Phase II | | | 1 (33.3) | 9 (45.0) | 2 (40.0) | 6 (46.2) | 7 (63.6) | 15 (88.2) | 6 (85.7) | 46 (58.2) |
| *S. dysenteriae* | 1 (3.3) | | | | 1 (1.9) | | | | | 2 (0.5) |
| *S. boydii* | | | 1 (3.3) | | 2 (3.8) | 1 (2.0) | | | 1 (3.2) | 5 (1.2) |
| Total | 30 (100.0) | 46 (100.0) | 30 (100.0) | 82 (100.0) | 53 (100.0) | 51 (100.0) | 46 (100.0) | 51 (100.0) | 31 (100.0) | 420 (100.0) |

[a]The proportion of *S. flexneri* and *S. sonnei* subspecies was calculated based on the total number of positive isolates for each respective species.
[b]In 2016, samples were collected from June to December only.

intermediate resistance to CRO, while *S. flexneri* showed 2% resistance. A large multi-center study in eight Asian countries from 2001 to 2004 also reported an increase in CRO resistance (5%) among *Shigella* isolates (30). These findings suggest that the most suitable antibiotic for treating *Shigella* infections may vary between *S. sonnei* and *S. flexneri*, with the former being more challenging to treat. The study found that resistance to SXT and NA was high from 2014 to 2017, while resistance to MEL was lower (2.8%).

The strengths of our study included an unbiased sampling approach that considered factors such as age, sex, nutrition status, disease severity, and socioeconomic context and samples from eight different geographical locations over a 9-year period. However, there are also several limitations. Due to funding constraints, the study was only conducted at 10 sentinel sites, which may not accurately reflect the full diversity of *Shigella* epidemiology and burden within the country. Moreover, the study was halted due to funding gap from January to May of 2016. A higher density surveillance network may provide more detailed insights. Additionally, the study only used conventional culture methods, not PCR, for confirming *Shigella* cases, which may have limited sensitivity (3), especially since around 46% of participants reported taking at least one dose of antibiotics for their current illness before enrollment. The use of quantitative PCR has revealed a higher burden of Shigella-attributable diarrhea in low-resource settings than previously recognized with culture-based diagnostics (32, 33). The study also did not perform genetic typing, which could have added valuable insight into the

**TABLE 5** Antimicrobial resistance profiles of *Shigella* isolates in Bangladesh from May 2014 to May 2022

| | *S. flexneri* (n = 185) | | | *S. sonnei* (n = 68) | | |
|---|---|---|---|---|---|---|
| Antimicrobial agents | S | I | R | S | I | R |
| Ampicillin | 100 (54.1%) | 0 | 85 (46%) | 23 (33.8%) | 0 | 45 (66.2%) |
| Azithromycin | 141 (76.3%) | 4 (2.2%) | 40 (21.7%) | 10 (14.7%) | 1 (1.5%) | 57 (83.8%) |
| Ciprofloxacin | 29 (15.7%) | 13 (7.1%) | 143 (77.3%) | 0 | 1 (1.5%) | 67 (98.5%) |
| Ceftriaxone | 179 (96.8%) | 2 (1.1%) | 4 (2.2%) | 29 (42.6%) | 25 (36.8%) | 14 (20.6%) |
| Mecillinam | 172 (93%) | 6 (3.3%) | 7 (3.8%) | 68 (100%) | 0 | 0 |
| Nalidixic acid | 11 (6%) | 2 (1.1%) | 172 (93%) | 0 | 0 | 68 (100%) |
| Trimethoprim/sulfamethoxazole | 100 (54.1%) | 0 | 85 (46%) | 9 (13.2%) | 9 (13.2%) | 50 (73.5%) |

TABLE 6   Multidrug resistance profiles of *Shigella* isolates in Bangladesh from May 2014 to May 2022[a]

| Antibiotic resistance | Number of isolates (%) | | MDR profile | S. flexneri n (%) | S. sonnei n (%) | Total |
|---|---|---|---|---|---|---|
| | S. flexneri, n = 182 | S. sonnei, n = 68 | | | | |
| ≥1 CLSI class | 182 | 68 | AMP-CIP-NA | 24 (96) | 1 (4) | 25 |
| ≥2 CLSI class | 162 | 66 | AMP-AZM-CIP-CRO-NA-SXT | 1 (3.6) | 27 (96.4) | 28 |
| ≥3 CLSI class | 112 | 60 | CIP-NA-SXT | 22 (91.7) | 2 (8.3) | 24 |
| ≥4 CLSI class | 58 | 50 | AMP-CIP-NA-SXT | 17 (100.0) | 0 | 17 |
| ≥5 CLSI class | 22 | 43 | AMP-AZM-CIP-NA-SXT | 15 (88.2) | 2 (11.8) | 17 |
| | | | AMP-AZM-CRO-NA-SXT | 2 (12.5) | 14 (87.5) | 16 |
| | | | AZM-CIP-NA-SXT | 8 (66.7) | 4 (33.3) | 12 |
| | | | AMP-AZM-CIP-NA | 6 (100.0) | 0 | 6 |
| | | | AZM-NA-SXT | 1 (14.3) | 6 (85.7) | 7 |
| | | | AZM-CIP-NA | 5 (83.3) | 1 (16.7) | 6 |
| | | | AZM-CIP-MEL-NA | 3 (100.0) | 0 | 3 |
| | | | AMP-AZM-CIP-MEL-NA | 3 (100.0) | 0 | 3 |
| | | | AMP-NA-SXT | 1 (50.0) | 1 (50.0) | 2 |
| | | | AMP-MEL-NA | 1 (100.0) | 0 | 1 |
| | | | AMP-CIP-MEL-NA | 1 (100.0) | 0 | 1 |
| | | | AMP-CIP-CRO-NA-SXT | 1 (100.0) | 0 | 1 |
| | | | AMP-AZM-NA-SXT | 1 (33.3) | 2 (66.7) | 3 |
| | | | AMP-AZM-CRO-NA | 0 | 1 (100.0) | 1 |
| | | | AMP-AZM-CIP-CRO-NA | 0 | 1 (100%) | 1 |

[a]CLSI, Clinical and Laboratory Standards Institute; MDR, multidrug resistant.

mechanisms of antibiotic resistance. The results of antibiotic resistance testing were only qualitative, not quantitative, and did not determine resistance levels that may be considered resistance. Lastly, there was no follow-up with patients after treatment or discharge, resulting in a lack of data on clinical outcomes and mortality.

In conclusion, our research indicated a noticeable rise in the prevalence of *S. sonnei* as the second most common *Shigella* species over the years. Despite having a high level of sensitivity to MEL, the *Shigella* strains in the study demonstrated resistance to antibiotics like AMP, AZM, NA, and SXT. The widespread resistance to fluoroquinolones and third-generation cephalosporins limits the available treatment options for shigellosis. This rising resistance to first-line antimicrobials emphasizes the need for new preventive and therapeutic measures. There are vaccines and alternative treatments in development, such as a live attenuated oral vaccine (WRSS1) against *S. sonnei* that has been found to be safe and effective in Bangladesh (34). Finally, ongoing monitoring of antibiotic drug susceptibility and determination of serotypes are necessary to comprehend the spread of *Shigella*.

The increasing resistance to fluoroquinolones and macrolides emphasizes the urgency for the development of a vaccine to prevent against *Shigella* infections. In order to plan for vaccine trials effectively and gain insights into the impact of *Shigella* infections in areas with limited resources, it is essential to have comprehensive data on *Shigella* infection epidemiology. Such surveillance is extremely important for preparation of preventive and vaccination efforts in Bangladesh and globally.

## ACKNOWLEDGMENTS

This work was supported by the Bill and Melinda Gates Foundation. We also thank the Fogarty International Center and National Institute of Allergy and Infectious Diseases, Training Grant in Vaccine Development and Public Health (D43 TW005572, AI155414, and AI177075). We are thankful to officials from the Government of Bangladesh and the participants who took part in interviews that provided support to this analysis. International Centre for Diarrhoeal Disease Research, Bangladesh, is grateful to the

governments of Bangladesh, Canada, Sweden, and the United Kingdom for providing core/unrestricted support.

## AUTHOR AFFILIATIONS

[1]International Centre for Diarrheal Disease Research Bangladesh, Dhaka, Bangladesh
[2]Institute of Epidemiology, Disease Control and Research (IEDCR), Dhaka, Bangladesh
[3]Department of Medicine, Infectious Diseases Division, Massachusetts General Hospital, Boston, Massachusetts, USA

## AUTHOR ORCIDs

Mokibul Hassan Afrad http://orcid.org/0000-0002-9627-7877
Taufiqur Rahman Bhuiyan https://orcid.org/0000-0003-3755-4763
Edward T. Ryan http://orcid.org/0009-0005-0169-394X
Firdausi Qadri http://orcid.org/0000-0002-8928-9888

## FUNDING

| Funder | Grant(s) | Author(s) |
| --- | --- | --- |
| Bill and Melinda Gates Foundation (GF) | | Mokibul Hassan Afrad |
| | | Taufiqul Islam |
| | | Yasmin Ara Begum |
| | | Md Saifullah |
| | | Faisal Ahmmed |
| | | Zahid Hasan Khan |
| | | Zakir Hossain Habib |
| | | Ahmed Nawsher Alam |
| | | Tahmina Shirin |
| | | Taufiqur Rahman Bhuiyan |

## ETHICS APPROVAL

All participants provided informed written consent, and for minors under 18 years old, written consent was obtained from their guardians. The study protocol was approved by the Research Review Committee and Ethical Review Committee of the International Centre for Diarrhoeal Disease Research, Bangladesh (icddr,b).

## ADDITIONAL FILES

The following material is available online.

Open Peer Review

**PEER REVIEW HISTORY (review-history.pdf).** An accounting of the reviewer comments and feedback.

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
