## [Reviewer comments · Microbiology Spectrum]

Microbiology Spectrum

Antibiotic resistance and serotype distribution of *Shigella* strains in Bangladesh over a period of 2014 to 2022: evidence from a nationwide hospital-based surveillance for cholera and other diarrheal diseases

Mokibul Hassan Afrad, Taufiqul Islam, Yasmin Begum, Md Saifullah, Faisal Ahmmed, Zahid Khan, Zakir Habib, Ahmed Alam, Tahmina Shirin, Taufiqur Bhuiyan, Edward Ryan, Ashraful Khan, and Firdausi Qadri

Corresponding Author(s): Firdausi Qadri, International Centre for Diarrhoeal Disease Research Bangladesh

Review Timeline:

Submission Date:	March 26, 2024
Editorial Decision:	May 30, 2024
Revision Received:	August 19, 2024
Editorial Decision:	August 29, 2024
Revision Received:	September 1, 2024
Accepted:	September 6, 2024

Editor: Siu-Kei Chow

Reviewer(s): The reviewers have opted to remain anonymous.

Transaction Report:

DOI: <https://doi.org/10.1128/spectrum.00739-24>

Re: Spectrum00739-24 (Antibiotic resistance and serotype distribution of Shigella strains in Bangladesh over a period of 2014 to 2022: evidence from a nationwide hospital-based surveillance for cholera and other diarrheal diseases)

Dear Dr. Firdausi Qadri:

Thank you for the privilege of reviewing your work. Below you will find my comments, instructions from the Spectrum editorial office, and the reviewer comments.

Revision Guidelines

Sincerely,
Siu-Kei Chow
Editor
Microbiology Spectrum

Reviewer #1 (Comments for the Author):

This manuscript reports surveillance of over 20 thousand stools from patients of all ages presenting with diarrheal illness to over 10 hospitals in Bangladesh between 2014 and 2022, through icddr,b. The stool was collected, along with a questionnaire, on up to four patients per day who met the inclusion criteria. The stool underwent centralized, standard culturing for Shigella spp. Antisera and monoclonal antibody testing for serotyping, and disk diffusion testing for antimicrobial susceptibility (only every five isolates for *S. flexneri*). 31,517 patients were enrolled, 24,357 stool specimens were available for culture. Over 50% were from children age 5 or less. 445 stool specimens were positive for Shigella spp. 79.6% were *S. flexneri*, 18.9% were *S. sonnei*, 1.2%

were *S. boydii*, and 0.04% were *S. dysenteriae*. The isolation rate of *S. sonnei* increased over the years, with the highest rate seen in 2021 (32.7%). The most prevalent *S. flexneri* serotype was 2a (79.7%) followed by 3a (11.4%). For *S. sonnei*, phase-I serotype was the most common from 2014 to 2019. 94% of *S. flexneri* and 100% of *S. sonnei* were resistant to nalidixic acid. 65% of *S. sonnei* and 6% of *S. flexneri* were resistant to ceftriaxone. 61.5% of *S. flexneri* isolates and 88.2% of *S. sonnei* isolates were multidrug-resistant.

The work is well done and provides important information on the epidemiology and impact of shigellosis in Bangladesh. Hopefully further work will be performed on the banked isolates to better understand their genetic traits.

-In the abstract, please clarify that the surveillance occurred in patients presenting with diarrheal illness or equivalent.

-Line 57: The results do not mention results from the questionnaire.

-Line 62: Is there a typo with the number here?

-Line 69: Is nalidixic acid still used clinically?

-Line 78: The abstract says 96%, not 99%.

-Line 114: Written or verbal?

-Line 251: Are there hypotheses as to why *S. sonnei* appears more adapted to higher income social structure?

-Line 287: A minor point, but direct comparison of resistance rates would be easier to follow.

-Line 296: Is 95% correct?

-Line 324: *Shigella* appears twice here, maybe a typo?

-Why is mecillinam so much more active than ceftriaxone, is it stability against ESBLs? Does this have implications on therapy?

-The vast majority of diarrheal patients had a negative culture. Can the authors discuss this in context to help the readers?

Reviewer #2 (Comments for the Author):

Thank you for the opportunity to review the manuscript titled "Antibiotic resistance and serotype distribution of *Shigella* strains in Bangladesh over a period of 2014 to 2022: Evidence from a nationwide hospital-based surveillance for cholera and other diarrheal diseases." The authors nicely describe a nationwide surveillance program to determine the prevalences, AST patterns and risk factors of *Shigella* infections in all ages. The study's value lies in its large sample size and wide range of ages, as well as its assessment of antibiotic resistance in the isolates. I found the manuscript easy to read and follow. While I don't have major concerns with the manuscript, I do have some comments to help improve it.

Comments:

Please review the entire manuscript for proper italics, capitalization and spelling of *Shigella* and the species for consistency.

Abstract:

o Upon reading the full text the difference between the 96% (Line 64) and over 99% (Line 78%) of resistant *S. flexneri* is apparent but in the abstract these values seem to conflict and are confusing. Suggest the authors clarify the percentages.

o Line 73: "evidence"

o Line 79-80: The data support the latter part of the statement "the findings provide..." but "provide crucial data for managing *Shigella* infection" might be a stretch to claim, since treatment and response in patients was not an outcome of the study or more details on patient management were not provided. Perhaps consider "provide crucial antimicrobial resistance patterns..."

Introduction: none

Methods:

o Add full collection period as described in the abstract

o The authors indicate a gap in funding that interrupted collection but this was not added as a limitation of the study. Since epi data is provided in Table 4 by year it should be added. Also consider a footnote to Table 4, to indicate that 2016 was not a full year of collection.

o Add the sources/manufacturers of the agar used in the culture techniques

o Line 171: correct spelling of *flexneri*

Results:

o Table 3: add spaces before parentheses, some p-values that are significant are bold while others are not.

o Table 4:

i. Please add denominators or the total for each year and confirm the percentages are correct. I found it difficult to determine how the percentages for *S. flexneri* isolates were derived.

ii. Correct capitalization for the species.

iii. See comment above regarding 2016 collection and addition of a footnote.

iv. Do the authors have a theory about why there were no isolates for *S. sonnei* recovered in 2015? If so please address in the

discussion.

o Table 5:

i. N for *S. sonnei* is 68 yet each row totals 26 isolates. Please correct or address the discrepancy.

ii. The *S. flexneri* columns are missing the %.

Discussion:

o Line 242-243: These two sentences seem to contradict each other, even though the first is addressing overall patients aged >5 presenting for surveillance samples compared to positive detections; please make it more clear. Perhaps "The highest 'positive' detection rate...".

o Line 288: "previous"

o Line 304-305: In regards to the lower sensitivity of culture compared to PCR, can the authors comment on whether patients received antibiotics prior to the sampling? If so was it a single dose or more?

o Please address how over-enrollment (Line 145) in an age group may have impacted the results of the study, particularly for the risk factors analysis.

POINT-BY-POINT RESPONSE TO REVIEWERS

We thank the Reviewers for their useful comments, which no doubt contributed to make this a better paper. In the following, a point-by-point response to all the questions and comments is provided. Moreover, we have corrected a few marginal percentage errors present in the previous manuscript in this revised version and apologize for those mistakes. We have updated **Table 4** as per the reviewer's advice. Additionally, **Table 5** has been corrected based on comment 14/I from Reviewer 2, and the manuscript has been revised accordingly. Furthermore, we have also revised **Table 1** to illustrate the division-wise detection rate of *Shigella* across Bangladesh and rephrased the relevant lines (233-237) in the discussion section. The original questions are in blue, our replies in black.

Moreover, we also received a suggestion from the Journal staff to add one more citation in the methods section. We did that accordingly.

Lines 129-130, The surveillance methods utilized in this study have been previously described in Khan Al et al. [15]. The procedure has been summarized below for the reader's convenience.

15. Khan, A.I., et al., *Epidemiology of cholera in Bangladesh: findings from nationwide hospital-based surveillance, 2014–2018*. *Clinical Infectious Diseases*, 2020. **71**(7): p. 1635-1642.

Items included in this re-submission:

- I. Cover letter.
- II. Point-by-point Response to reviewers (Response to Reviewers.docx),
- III. Revised manuscript marked-up copy (Manuscript_markedup.docx)
- IV. Revised manuscript clean copy (Manuscript.docx).
- V. Revised tables: Table 1, 3, 4, and 5

REVIEWER COMMENTS

REVIEWER #1

This manuscript reports surveillance of over 20 thousand stools from patients of all ages presenting with diarrheal illness to over 10 hospitals in Bangladesh between 2014 and 2022, through icddr,b. The stool was collected, along with a questionnaire, on up to four patients per day who met the inclusion criteria. The stool underwent centralized, standard culturing for *Shigella* spp. Antisera and monoclonal antibody testing for serotyping, and disk diffusion testing for antimicrobial susceptibility (only every five isolates for *S. flexneri*). 31,517 patients were enrolled, 24,357 stool specimens were available for culture. Over 50% were from children age 5 or less. 445 stool specimens were positive for *Shigella* spp. 79.6% were *S. flexneri*, 18.9% were *S. sonnei*, 1.2% were *S. boydii*, and 0.04% were *S. dysenteriae*. The isolation rate of *S. sonnei* increased over the years, with the highest rate seen in 2021 (32.7%). The most prevalent *S. flexneri* serotype was 2a (79.7%) followed by 3a (11.4%). For *S. sonnei*, phase-I serotype was the most common from 2014 to 2019. 94% of *S. flexneri* and 100% of *S. sonnei* were resistant to nalidixic acid. 65% of *S. sonnei* and 6% of *S. flexneri* were resistant to ceftriaxone. 61.5% of *S. flexneri* isolates and 88.2% of *S. sonnei* isolates were multidrug-resistant.

The work is well done and provides important information on the epidemiology and impact of shigellosis in Bangladesh. Hopefully further work will be performed on the banked isolates to better understand their genetic traits.

We thank the reviewer for acknowledging the interest of this manuscript and for appreciating our analysis.

In the abstract, please clarify that the surveillance occurred in patients presenting with diarrheal illness or equivalent.

We have revised and included as the reviewer suggested.

Lines 54-55, "Stool specimens were collected from patients with acute watery diarrhea and tested for *Shigella* species by microbiological culture."

Line 57: The results do not mention results from the questionnaire.

Due to word limitations, the authors did not mention the questionnaire findings in the abstract. However, these findings are detailed in the results section of the manuscript (lines 185-190).

Line 62: Is there a typo with the number here?

Corrected.

Line 70: Is nalidixic acid still used clinically?

Thanks to the reviewer for pointing this. Nalidixic acid is not being used clinically, therefore we have rephrased line 70,

"The high level of resistance to ciprofloxacin highlights the need for more prudent use of this antibiotic and improved hygiene and sanitation."

Line 78: The abstract says 96%, not 99%.

Corrected (line 78).

Line 114: Written or verbal?

Consent was obtained as written (line 113).

Line 251: Are there hypotheses as to why *S. sonnei* appears more adapted to higher income social structure?

The authors are unable to comment due to the lack of supportive literature. However, we have cited the relevant literature available.

Line 389, Ref 20: Thompson, C.N., P.T. Duy, and S. Baker, The Rising Dominance of Shigella sonnei: An Intercontinental Shift in the Etiology of Bacillary Dysentery. PLoS Negl Trop Dis, 2015. 9(6): p. e0003708.

Line 287: A minor point, but direct comparison of resistance rates would be easier to follow.

Thank you and we have revised as suggested.

*Lines 281-282, "The *S. sonnei* strains also showed a high resistance (66.2%) to AMP, while *S. flexneri* showed 46% resistance to AMP."*

Line 296: Is 95%, correct?

We apologize for the mistake. The actual percent resistance is 2.8% (7/253) to MEL. Now corrected (line 291),

"The study found that resistance to SXT and NA was high from 2014 to 2017, while resistance to MEL was lower (2.8%)."

Line 324: Shigella appears twice here, maybe a typo?

Corrected.

Why is mecillinam so much more active than ceftriaxone, is it stability against ESBLs? Does this have implications on therapy?

Mecillinam's higher efficacy in Bangladesh is attributed to its stability against extended-spectrum beta-lactamases (ESBLs), which typically confer resistance to many beta-lactam antibiotics. Its unique mechanism of action and stability against ESBLs make mecillinam effective against these resistant strains. Additionally, its less frequent use may also contribute to its greater effectiveness.

The vast majority of diarrheal patients had a negative culture. Can the authors discuss this in context to help the readers?

Thank you to the reviewer for highlighting this critical issue. The sensitivity of the culture-based method is lower compared to PCR, and this study relies solely on the culture-based method. Additionally, our dataset (not included in the manuscript) indicates that approximately 46% of participants had taken antibiotics for their current illness before enrollment. We have included this in the limitations section of the manuscript.

*Lines 299-302, "Additionally, the study only used conventional culture methods, not PCR, for confirming *Shigella* cases, which may have limited sensitivity [3], especially since around 46% of participants reported taking at least one dose of antibiotics for their current illness before enrollment."*

REVIEWER #2

Thank you for the opportunity to review the manuscript titled "Antibiotic resistance and serotype distribution of Shigella strains in Bangladesh over a period of 2014 to 2022: Evidence from a nationwide hospital-based surveillance for cholera and other diarrheal diseases." The authors nicely describe a nationwide surveillance program to determine the prevalences, AST patterns and risk factors of Shigella infections in all ages. The study's value lies in its large sample size and wide range of ages, as well as its assessment of antibiotic resistance in the isolates. I found the manuscript easy to read and follow. While I don't have major concerns with the manuscript, I do have some comments to help improve it.

We thank the reviewer for appreciating our analysis and suggestions provided to improve the paper.

Please review the entire manuscript for proper italics, capitalization and spelling of Shigella and the species for consistency.

We have revised and corrected.

ABSTRACT

Upon reading the full text the difference between the 96% (Line 64) and over 99% (Line 78) of resistant S. flexneri is apparent but in the abstract these values seem to conflict and are confusing. Suggest the authors clarify the percentages.

Corrected (*line 78*).

Line 73: "evidence"

Corrected (*line 73*).

Line 79-80: The data support the latter part of the statement "the findings provide..." but "provide crucial data for managing Shigella infection" might be a stretch to claim, since treatment and response in patients was not an outcome of the study or more details on patient management were not provided. Perhaps consider "provide crucial antimicrobial resistance patterns..."

Thanks to the reviewer and we have rephrased the line.

Now lines 79-80 reads, "**The findings provide crucial antimicrobial resistance patterns of Shigella species**, highlighting the need for ongoing resistance monitoring and potentially informing future vaccine trials."

METHODS

Add full collection period as described in the abstract

Added. *Lines 125-126*, "**In this study, participants were enrolled from May 2014 to May 2022.**"

The authors indicate a gap in funding that interrupted collection but this was not added as a limitation of the study. Since epi data is provided in Table 4 by year it should be added. Also consider a footnote to Table 4, to indicate that 2016 was not a full year of collection.

As the reviewer suggested, we have added this as a limitation (lines 300-301), we have also added a footnote in Table 4 as the reviewer suggested.

Lines 297-298, "**Moreover, the study was halted due to funding gap from January to May of 2016.**"

Table 4 footnote, "**** In 2016, samples were collected from June to December only.**"

Add the sources/manufacturers of the agar used in the culture techniques

Added (*lines 154 and 156*).

“... .. streaking directly on Salmonella Shigella (SS) agar (Becton Dickinson, France) and incubated at 37 °C for 16-22 h. The resultant Shigella-like colonies were picked onto Kligler iron agar (KIA) Motility Indole Urea (MIU), Simmon citrate agar (Becton Dickinson, France)”

Line 171: correct spelling of flexneri

Corrected (*line 167*).

RESULTS

Table 3: add spaces before parentheses, some p-values that are significant are bold while others are not.

Corrected.

Table 4

I. Please add denominators or the total for each year and confirm the percentages are correct. I found it difficult to determine how the percentages for *S. flexneri* isolates were derived.

We have added the total number for each year in table 4. In the revised table 4, the proportion of *S. flexneri* and *S. sonnei* sub-species was calculated based on the total number of positive isolates for each respective species.

II. Correct capitalization for the species.

Corrected.

III. See comment above regarding 2016 collection and addition of a footnote.

We have added the footnote.

IV. Do the authors have a theory about why there were no isolates for *S. sonnei* recovered in 2015? If so please address in the discussion.

We followed the same laboratory procedure throughout the study. Therefore, we cannot explain why no *S. sonnei* was detected in 2015.

Table 5

I. N for *S. sonnei* is 68 yet each row totals 26 isolates. Please correct or address the discrepancy.

We are sorry for the mistake. The N for *S. sonnei* is 68 and we have revised Table 5 accordingly.

II. The *S. flexneri* columns are missing the %.

Corrected.

DISCUSSION

Line 242-243: These two sentences seem to contradict each other, even though the first is addressing overall patients aged >5 presenting for surveillance samples compared to positive detections; please make it more clear. Perhaps "The highest 'positive' detection rate...".

We thank the reviewer and have revised as suggested.

Lines 239, “The highest positive detection rate was observed in children aged 6-17 years.

Line 288: "previous"

Spelling corrected (*line 283*).

Line 304-305: In regard to the lower sensitivity of culture compared to PCR, can the authors comment on whether patients received antibiotics prior to the sampling? If so was it a single dose or more?

Thank you to the reviewer for highlighting this critical issue. Our dataset shows that approximately 46% of participants had taken antibiotics for their current illness before enrollment. We have added this in the manuscript.

Lines 300-302, "Additionally, the study only used conventional culture methods, not PCR, for confirming *Shigella* cases, which may have limited sensitivity [3], especially since around 46% of participants reported taking at least one dose of antibiotics for their current illness before enrollment."

Please address how over-enrollment (Line 145) in an age group may have impacted the results of the study, particularly for the risk factors analysis.

The study aimed to enroll at least 4 participants each day, 5 days a week. However, due to low patient numbers on some days, our standard practice is to enroll 20 participants per week. Although there were instances of over-enrollment on certain days, these were balanced out once the weekly target was achieved.

Re: Spectrum00739-24R1 (Antibiotic resistance and serotype distribution of Shigella strains in Bangladesh over a period of 2014 to 2022: evidence from a nationwide hospital-based surveillance for cholera and other diarrheal diseases)

Dear Dr. Firdausi Qadri:

Thank you for the privilege of reviewing your work. Below you will find my comments, instructions from the Spectrum editorial office, and the reviewer comments.

Revision Guidelines

Sincerely,
Siu-Kei Chow
Editor
Microbiology Spectrum

Reviewer #1 (Comments for the Author):

- Line 166: whisch -> which
- Line 179: Significance is usually defined by p values <0.05, not =<0.05, unless there is some special intention here.
- Line 220: higher resistance rates
- Line 235: Is there anything special about Rangpur Division?
- Line 245: Remove the bracket.
- Line 257: may become a

POINT-BY-POINT RESPONSE TO REVIEWERS

We thank the Reviewers for his/her comments.

In the following, a point-by-point response to all the questions and comments is provided. The original questions are in blue, our replies in black.

Items included in this re-submission:

- I. Cover letter
- II. Point-by-point Response to reviewers (Response to Reviewers.docx)
- III. Revised manuscript marked-up copy (Manuscript_markedup.docx)
- IV. Revised manuscript clean copy (Manuscript.docx).

REVIEWER COMMENTS

REVIEWER #1

Line 166: *whisch* -> *which*

Corrected (line 166).

Line 179: Significance is usually defined by p values <0.05, not =<0.05, unless there is some special intention here.

Thanks to the reviewer for pointing the issue. Now corrected (line 179).

Line 220: *higher resistance rates*

Thanks to the reviewer. Now line 220 reads,

“Meanwhile, *S. sonnei* isolates showed **higher resistance rates** to a variety of antibiotics, including 100% resistance to NA, followed by 98.5% to CIP, 83.8% to AZM, 73.5% to SXT, 66.2% to AMP, and 20.6% to CRO.”

Line 235: *Is there anything special about Rangpur Division?*

The sentinel site in the Rangpur Division is Thaurgaon district, located in the northwest of the country (Figure 1). Historically, the prevalence of enteric pathogens in this area has been consistently low.

Line 245: *Remove the bracket.*

Corrected (line 245).

Line 257: *may become a*

Thanks to the reviewer. Now line 257 reads,

“As Bangladesh continues to develop economically and improve its sanitation, *S. sonnei* **may become a** more significant public health concern in the future.”

Re: Spectrum00739-24R2 (Antibiotic resistance and serotype distribution of Shigella strains in Bangladesh over a period of 2014 to 2022: evidence from a nationwide hospital-based surveillance for cholera and other diarrheal diseases)

Dear Dr. Firdausi Qadri:

Your manuscript has been accepted, and I am forwarding it to the ASM production staff for publication. Your paper will first be checked to make sure all elements meet the technical requirements. ASM staff will contact you if anything needs to be revised before copyediting and production can begin. Otherwise, you will be notified when your proofs are ready to be viewed.

Sincerely,
Siu-Kei Chow
Editor
Microbiology Spectrum